# Influence of the Mix Proportion and Aggregate Features on the Performance of Eco-Efficient Fine Recycled Concrete Aggregate Mixtures

**DOI:** 10.3390/ma15041355

**Published:** 2022-02-12

**Authors:** Diego Jesus De Souza, Mayra T. de Grazia, Hian F. Macedo, Leandro F. M. Sanchez, Gabriella P. de Andrade, Olga Naboka, Gholamreza Fathifazl, Pierre-Claver Nkinamubanzi

**Affiliations:** 1Department of Civil Engineering, Faculty of Engineering, University of Ottawa, Ottawa, ON K1N 6N5, Canada; mtagl010@uottawa.ca (M.T.d.G.); hdefr095@uottawa.ca (H.F.M.); leandro.sanchez@uottawa.ca (L.F.M.S.); puentedg@gmail.com (G.P.d.A.); 2National Research Council Canada, Ottawa, ON K1V 1J8, Canada; olga.naboka@nrc-cnrc.gc.ca (O.N.); reza.fathifazl@nrc-cnrc.gc.ca (G.F.); pierre-claver.nkinamubanzi@nrc-cnrc.gc.ca (P.-C.N.)

**Keywords:** fine recycled concrete aggregate, residual cement paste (RCP), mix design technique of concrete, rheological behaviour, hardened property of concrete

## Abstract

Most of the previous research on recycled concrete aggregates (RCA) has focused on coarse RCA (CRCA), while much less has been accomplished on the use of fine RCA particles (FRCA). Furthermore, most RCA research disregards its unique microstructure, and thus the inferior performance of concrete incorporating RCA is often reported in the fresh and hardened states. To improve the overall behaviour of RCA concrete advanced mix design techniques such as equivalent volume (EV) or particle packing models (PPMs) may be used. However, the efficiency of these procedures to proportion eco-efficient FRCA concrete still requires further investigation. This work evaluates the overall fresh (i.e., slump and rheological characterization) and hardened states (i.e., non-destructive tests, compressive strength and microscopy) performance of sustainable FRCA mixtures proportioned through distinct techniques (i.e., direct replacement, EV and PPMs) and incorporating different types of aggregates (i.e., natural and manufactured sand) and manufacturing processes (i.e., crusher fines and fully ground). Results demonstrate that the aggregate type and crushing process may influence the FRCA particles’ features. Yet, the use of advanced mix design techniques, particularly PPMs, may provide FRCA mixes with quite suitable performance in the fresh (i.e., 49% lower yield stress) and hardened states (i.e., 53% higher compressive strength) along with a low carbon footprint.

## 1. Introduction

Concrete is a widely used material for critical infrastructure worldwide; over 10 billion tons of concrete are produced annually in the modern industrial society [1]. However, the construction industry faces crucial challenges in finding cost-effective strategies to reduce the carbon footprint and embodied energy associated with producing and using concrete and its ingredients [2,3,4]. Amongst the viable solutions, the re-use of construction waste, the so-called recycled concrete aggregates (RCA), has been receiving increased attention in the civil industry [5,6,7,8,9,10].

The intrinsic difference between RCA and natural aggregate (NA) is the fact that CRCA is a two-phase material composed of the original virgin aggregate (OVA) and residual mortar (RM). Likewise, fine recycled concrete aggregates (FRCA) comprise the original virgin aggregate (OVA) and residual cement paste (RCP). The properties of RCA concrete made of CRCA have been extensively investigated [3,10,11,12,13,14,15,16,17]; therefore, numerous documents such as ACI-555R [18], BS EN 1744-6:2006 and BS EN 933-11:2009 along with RILEM technical reports, such as [19,20], provide guidelines and criteria on how to deal with and use CRCA for proportioning RCA concrete. Yet, although RCA’s use in concrete has grown over the past years, concerns related to its mechanical properties, durability and long-term behaviour prevent its widespread use for structural purposes [6,9,14,21,22], especially due to the lack of efficient quality control and proportioning procedures. On the other hand, FRCA is considered a very low-quality material due to its high amount of RCP adhered to the fine particles [15,21]; thus, much less has been developed and accomplished with the use of FRCA. The amount of RCP varies depending upon the type and quality of the OVA (i.e., lithotype, texture, shape, etc.), RCP inner quality (i.e., mechanical properties), crusher type and manufacturing process, etc., which leads to substantial variability to the final product [13,23,24]. The few existing studies suggest that FRCA negatively influences the fresh state, mechanical properties and durability of recycled concrete [22]. Otherwise, it has been demonstrated that FRCA replacement ratios lower than 30% do not negatively impact the mechanical properties of recycled concrete [23].

Studies have demonstrated that mortar mixtures made of partial or full direct replacement of NA by FRCA displayed more somewhat similar fresh state behaviour than conventional mortar, once both aggregates (either NA or FRCA) were in saturated surface-dry conditions [24]. Conversely, it has been verified that self-consolidating recycled concrete mixtures incorporating 50% and 100% of FRCA demonstrated less self-compacting ability than conventional mixtures [25]. Moreover, results suggest that recycled mixtures bearing 100% of FRCA may yield 35% higher drying shrinkage [26] along with 30% and 50% lower compressive strength [27] and modulus of elasticity [28,29], respectively, than conventional companion mixtures. In all the above studies, the low performances of the recycled mixtures were deemed to be linked to some of the FRCA features, such as their higher porosity, absorption and the presence of important amounts of residual cement paste adhered to the particles.

In this context, advanced mix design techniques have been developed to optimize recycled concrete mixtures, allowing more efficient re-use of RCA in concrete and enabling better performances in the fresh and hardened states. The literature suggests that the key factor for an efficient proportioning of recycled concrete mixtures bearing CRCA is to account for their unique microstructure (i.e., original virgin aggregate—OVA and residual mortar—RM), while the mix design and proportioning techniques such as the equivalent mortar volume (EMV) [30] and equivalent volume (EV) [31] methods were established with this concept and showed promising results. Likewise, particle packing models (PPMs) have demonstrated to successfully proportion recycled mixtures bearing CRCA with suitable fresh- and hardened-state properties [32]. Nevertheless, to the authors’ best knowledge, the literature has not largely explored mix design techniques accounting for the unique FRCA microstructure. Thus, this work aims to appraise the influence of the FRCA features and mix-proportion technique on the fresh and hardened performance of sustainable recycled concrete mixtures designed for structural purposes.

## 2. Background in Mix Proportioning Techniques

### 2.1. Proportion RCA Concrete

#### 2.1.1. Direct Replacement Methods

One of the earliest attempts to use a mix proportion of RCA concrete was performed through direct replacement methods (DRMs), which can be conducted by weight or volume. This approach treats RCA as homogeneous material and partially replaces a certain amount of natural aggregates accounting for their distinct microstructure and or the presence of RM [33,34]. Moreover, the direct replacement of the NA by RCA (i.e., OVA + RM) without accounting for the RM makes the total amount of coarse aggregate (i.e., OVA and NA) lower than the NA in the companion conventional mixture. Thus, DRM techniques often result in recycled concrete mixtures with similar fresh-state behaviour. Still, it is quite inferior hardened-state performance compared to a companion CC mix, especially when moderate to high amounts of RCA is used [35]. Nevertheless, recent studies have shown that suitable fresh- and hardened-state performances of RCA concrete might be achieved using DRM when adjustments are made to the water-to-binder ratio along with high amounts of binder materials (e.g., >400 kg/m^3^) and chemical admixtures [36,37]. In other words, this approach offsets the use of RCA since PC is by far the concrete ingredient presenting the most negative impact on sustainability [38]. Hence, “equivalent volume methods” were proposed in the literature, and amongst those, the equivalent volume (EV) method stands out as a promising technique.

#### 2.1.2. Equivalent Volume Methods

The equivalent mortar volume (EMV) method was initially proposed by Fathifazl et al. 2009a and treated RCA as a multi-phase material composed of RM and OVA. Thus, the volume of cementitious materials added to the recycled mix is reduced according to the amount of existing RM content adhered to the RCA particles. It has been suggested that EMV can reach similar or even somewhat higher compressive and tensile splitting strengths than conventional concrete [39,40]. Besides the comparable compressive strength, EMV-proportioned recycled mixtures often yield a similar modulus of elasticity than companion CC (i.e., varying, on average, from −5% to 11%) [41,42]. The enhancement in the modulus of elasticity of EMV-proportioned RCA concrete is due to the equivalent volumetric ratio of natural aggregate (i.e., either OVA or NA) between CC and the recycled mixture, whereas in DRM-proportioned RCA concrete, the overall volume of natural aggregate is decreased in the mixture. Otherwise, EMV-proportioned mixtures often face challenges in the fresh state (e.g., high consistency, low flowability and unsuitable rheological profile) due to the lower availability of fresh mortar in the mixture [43]. Moreover, the RM was found to behave as an aggregate in the fresh state while acting as a mortar in the hardened state, negatively affecting the flowability of recycled mixtures [44,45].

In order to address the issues often faced in the fresh state by EMV-proportioned RCA mixes, especially in eco-efficient mixtures with a reduced binder content, a modification to the preliminary method was proposed by Hayles et al. 2018. The EMV-mod method adds a supplementary factor to the previous strategy to optimize the “cement-to-sand mass ratio” of the RM and thus allows the selection of the amount of cement to be accounted for in the new RCA mix. Hence, EMV-mod-proportioned mixtures show better improvement in fresh-state performance (i.e., lower consistency and higher flowability) than EMV mixes. However, this approach was verified to present an important drawback since the binder content is typically increased for a given targeted strength compared to conventional and EMV-designed mixes [44].

Recently, a new mix design procedure, the so-called equivalent volume (EV) method, was developed by Ahimoghadam et al. 2020 based on the main concepts of the previous EMV and EMV-mod methods. The EV was established to solve the issues faced by EMV and EMV-mod mix-proportioned mixtures in the fresh state, along with providing recycled mixtures with an eco-friendly character and suitable performance in the hardened state [31]. The novelty of this method is that the RCA-proportioned mix is based on a companion conventional concrete displaying the same volumetric amount of cement paste and aggregates (fine and coarse), while in the EMV (and EMV-mod), the same amount of coarse aggregates and mortar was considered for calculations. Furthermore, the RM is considered in this method as the summation of residual paste and residual fine aggregates. Hence, the total volumetric cement paste content in an EV-proportioned mixture is defined as the summation of residual and fresh-paste volumes. As opposed to EMV and EMV-mod methods, EV-proportioned mixes often yielded suitable fresh-state behaviour and cement efficiency (PC contents of about 300 kg/m^3^) due to the significant increase in fine aggregates and decrease in coarse particles in the system [31,41].

### 2.2. Particle Packing Models (PPM)

Particle packing models (PPMs) are advanced mix design techniques used to improve the performance of concrete mixtures in fresh and hardened states. PPMs are divided into discrete and continuous models; either discrete or continuous, PPMs aim to optimize granular systems’ particle size distribution (PSD), reducing the material’s porosity and consequently increasing the system’s packing density [46]. The discrete approach refers to the packing of multimodal distributions containing two or more discrete size classes of particles [47]. In contrast, the continuous model is based on the basic assumption that all possible aggregates are present within the particle distribution of the system [48,49]. Typically, PPM-mix-proportioned mixes present lower porosity and amounts of the binder when compared to conventionally designed concrete.

The first continuous PPM was established by Fuller in 1907, and it is based on a power curve that accounts for two variables: the distribution factor (*q*) and the largest particle size (*D*) [50]. Over the years, several authors have developed different models; yet, the newest and most recognized continuous PPM is the Alfred model (also known as modified Andreasen), which Funk and Dinger initially created in 1980. This model calculates the optimum PSD based on a coefficient of distribution (*q*) along with the largest (*D_L_*) and smallest (*D_S_*) particle sizes (i.e., diameter) present in the system (Equation (1)). The coefficient of distribution (*q*) is adopted based on the fresh-state requirements of the mix; values ranging from 0.20 to 0.23 are often selected for self-consolidating mixtures, whereas distribution factors of 0.26–0.28 are normally targeted for vibrated and or pumped concrete. It is worth noting that 0.37 is the q value that yields the highest packing density, hence the lowest porosity to granular systems [51].
(1)$$CPFT=100\times \left(\frac{D_{P}{}^{q}-D_{S}{}^{q}}{D_{L}{}^{q}-D_{S}{}^{q}}\right)$$
where: CPFT is the cumulative (volume) percent finer than *D_P_*, and *D_P_* is the particle size.

PPMs have been widely and successfully used to improve performance, especially in the hardened state of eco-efficient [32,52], conventional [52] and high-performance concrete [53,54]. Nevertheless, very little research has been conducted on the use of continuous PPMs to optimize FRCA concrete, especially to proportion eco-efficient recycled mixtures with full replacement of natural aggregates by FRCA [33,46].

## 3. Scope of Work

As previously described, there is currently limited research on the influence of the FRCA particle features (i.e., amount of residual cement paste, FRCA type: natural vs. manufactured and manufacturing process) and mix design techniques (i.e., conventional vs. advanced) on the fresh- and hardened-state performance of eco-efficient recycled concrete mixtures made of 100% FRCA. This work is divided into two main phases: (1) production and characterization of FRCA materials and (2) appraisal of the fresh- and hardened-state properties of eco-efficient FRCA concrete mixtures. The first phase evaluates the impact of the FRCA type (i.e., originally fabricated with natural vs. manufactured sand) and manufacturing process (i.e., crushing process) on the physical properties of the granular material. In the second phase, three mix design techniques (i.e., DRM, EV and PPM) are selected to proportion eco-efficient FRCA mixtures made of natural or manufactured sand. Finally, evaluations on the fresh- and hardened-state properties are conducted, and comparisons with a companion conventional concrete (0% FRCA) are performed. It is worth noting that the term eco-efficient used in the context of this work refers to concrete mixtures displaying equal or lower binder contents than their companion conventional mixtures. In other words, there is no need to use a supplementary amount of binder to obtain a performance in the recycled mixture as is conventionally adopted in RCA research.

## 4. Materials and Methods

The experimental program is divided into two main streams as described in Section 3 (i.e., FRCA characterization and appraisal of mix design techniques). However, the FRCA materials’ characterization (i.e., production and characterization of FRCA materials) is displayed in this section since they are initially required to properly create a mix-proportion of recycled concrete mixtures for analyses in the fresh and hardened states.

### 4.1. FRCA Production Process

The FRCA used in this work was manufactured in the laboratory; it is derived from crushed companion conventional concrete specimens, mix-proportioned through the ACI method (i.e., absolute volume method) and displaying a design compressive strength (i.e., 28-day value) of 35 MPa. Two families of FRCA were fabricated, one containing a natural siliceous sand (ACI-NS) and the other incorporating a limestone-manufactured sand (ACI-MS). A coarse limestone aggregate with a nominal maximum size of 19 mm, an oven-dry specific gravity of 2.78 g/cm^3^ and 0.75% absorption was used to produce the conventional concrete mixtures. Neither chemical admixture nor mineral admixtures were employed in the conventional concrete mixtures. Table 1 summarizes the conventional concrete mixture proportions used to fabricate the FRCA material.

Two hundred conventional concrete cylinders (i.e., 100 mm × 200 mm in size) were cast from each of the concrete mixtures used in this research incorporating NA and MS. The specimens were kept in the moulds at laboratory temperature, demoulded after 24 h and moist-cured over 28 days. In order to appraise the influence of the crushing process on the FRCAs’ inner quality, two manufacturing methods (i.e., crusher fines—CF and fully ground—FG) were employed to produce the recycled granular material. Figure 1 presents the CF method, where coarse and fine RCA are produced simultaneously. The CF process presents two series of crushing and three stages. First, concrete specimens were crushed into a 19 mm maximum gap opening in the jaw crusher. Then, the produced material followed a second crushing stage with a 5 mm gap opening. Finally, the material was divided into FRCA and CRCA; only the fine fraction (i.e., 150 μm–5 mm) was sieved and used in this work as per the Canadian Standards Association (CSA, [55]).

The second method (FG—full process Figure 1) only produces FRCA by multiples (and continuous) series of crushing (5 mm gap opening in the jaw crusher) and sieving as per CSA [55]. In both manufacturing methods, the particle size distribution of the final FRCA materials ranged from 150 µm to 5 mm. Furthermore, regardless of the crushing process, two FRCA mixtures were produced in this work: FRCA from natural (FRCA NS) and manufactured (FRCA MS) sands. Both were fabricated according to the two methods described above (i.e., CF and FG).

### 4.2. Materials Characterization

#### 4.2.1. FRCA


**
*Residual cement paste*
*(RCP)*
**


The RCP was measured in this work on all FRCA materials through the soluble silica sub-procedure according to ASTM C1084-15 [56] and C114-18 [57]. The procedure involves the selective extraction of silica from Portland cement by cold dilute hydrochloric acid with subsequent gravimetric analysis of soluble silica. The procedure assumes that only silica from Portland cement is soluble in cold dilute hydrochloric acid, while silica from aggregates is not soluble. Samples of 2.5 g of all FRCA materials were analyzed as per the above-mentioned procedures. They were then weighed to determine the mass difference before and after the test, representing the amount of residual cement paste in the FRCA material (Table 2). Overall, both the distinct raw materials (i.e., NS or MS) and manufacturing processes (i.e., CF or FG) influenced the amount of RCP adhered to the FRCA particles. Yet, the manufacturing process seems to have a greater impact on the RCP adhered to the FRCA particles. The RCP values for the various mixtures ranged from 11.4% (FRCA-MS-FG) to 16.8% (FRCA-MS-CF).


**
*Potential of RCP for further hydration*
**


Isothermal heat conduction calorimetry (IHCC) of the FRCA was determined in an isothermal calorimeter G-563-62-L0071/AD by Controls Group. The test was performed to determine whether the residual cement paste (RCP) adhered to the FRCA particles might have some potential for further hydration in concrete. The test was conducted on original FRCA particles and cement powder for comparison purposes. It is worth noting that IHCC was performed on freshly prepared samples to avoid the possible reaction of unhydrated cement with moisture from the air. Ten grams of each FRCA material were selected for testing; 3.5 g of double-deionized water was added to the samples to evaluate the cementing capability of the RCP encountered in the FRCA particles. After mixing for 60 s, the samples were placed in the calorimeter and tested. It is important to note that the plastic scintillation vials were filled with 26.49 ± 0.2 g of glass beads as a reference. All the IHCC results are presented on the same basis (per gram of cement) since the amount of cement in each sample is variable. The IHCC results obtained in this work are presented in the analysis and discussion section.


**
*Specific gravity and water absorption*
**


The specific gravity and water absorption capacity of all FRCA materials were evaluated according to the method proposed by [58]. The conventional procedure from CSA [55] was found to not yield reliable results for FRCA due to its cohesiveness and binding behaviour. As per [58], the samples were first saturated in 0.1% of sodium hexametaphosphate solution, a clay dispersant commonly used in soil analysis for clay suspensions. Then, for a given sample, a concentration of 1 g/L of the dispersant was used to prevent the agglomeration of the particles. Then, the procedure was performed as per CSA [55], except that the samples were left for a further 24 h in a pycnometer to have entrapped air bubbles removed, as proposed by [58]. Table 2 summarizes the FRCAs’ physical characterization results and the properties of natural and manufactured fine aggregates [55].


**
*Particle size distribution (PSD)*
**


Figure 2a shows the particle size distribution (PSD) for the FRCA materials derived from natural sand (i.e., NS-CF and NS-FG), while Figure 2b illustrates the results of the FRCA materials derived from manufactured sand (i.e., MS-CF and MS-FG). The square-dotted lines represent the particle size distribution thresholds of CSA [55]. Overall, the PSD of the FRCA can be significantly affected by the crushing process adopted. For instance, FRCA-NS-CF (crusher fines—Figure 2a) predominantly comprises coarser particles, yielding a fineness modulus of 3.27, whereas FRCA-NS-FG (fully ground—Figure 2a) displayed a higher amount of fine particles and thus a fineness modulus of 2.53. Moreover, FRCA-NS-FG presented a PSD similar to the natural sand and fineness modulus. The same trend was observed for the FRCA-MS from crusher fines and fully ground types (Figure 2b). FRCA-MS-CF presented a fineness modulus of 3.17, while FRCA MS-FG and the original manufactured sand displayed fineness moduli values of 2.70 and 2.85, respectively.

#### 4.2.2. Binder and Filler

A general-use (equivalent to ASTM Type I) Portland cement was used in all mixes. In addition, a limestone filler with a smaller PSD than PC (Figure 3), known as performance filler, was employed in the mixtures proportioned through PPM methods. The chemical compositions of the PC and filler, obtained via X-ray fluorescence in a Rigaku Supermini200 WDXRF Spectrometer, and the PC mineralogical phases calculated through Bogue’s equations [59] are presented in Table 3. Moreover, the specific gravity and specific surface area analyses were performed through a gas pycnometer (AccuPyc II by Micromeritics) and BET (Horiba SA-9600 by Meritics), respectively, and the results are summarized in Figure 3.

### 4.3. Concrete Manufacturing

#### 4.3.1. Mix design Approaches (DRM, EV and PPM)

A total of three distinct concrete mixtures (i.e., DRM, EV and PPM) with 100% replacement of fine NA by FRCA were designed. Moreover, two control mixes (e.g., ACI method—0% FRCA) containing NS or MS were also produced for comparison purposes. Concrete mixtures proportioned using the DRM method were designed according to the ACI protocol (i.e., absolute volume method), purposely treating FRCA as presenting the same properties of fine NA; the latter is a common practice adopted in the construction industry.

The EV method [31] was adapted to design recycled mixtures made of 100% FRCA and accounting for the RCP content attached to the fine aggregate particles. However, the main concept of the EV technique was kept the same as follows: the proportioned FRCA mixture is based on a conventional mixture, having the same amount (in volume) of cement paste and aggregates. In other words, the total amount of cement paste in the EV mix is considered as the summation of the RCP plus the fresh paste added to the mix. In contrast, the total volume of aggregates is the volume of original virgin aggregates (OVA) plus the volume of natural aggregates (NA). In addition to the above methods, a novel approach incorporating the EV concept (i.e., accounting for the RCP of the FRCA) into a continuous PPM technique (i.e., Alfred’s model) was also adopted in this research. A limestone filler was used in the PPM-mix-designed mixtures to lessen the overall interparticle porosity and binder content of the system. The distribution factor (q) selected in this work was 0.28 to ensure a suitable fresh-state behaviour (i.e., targeting a vibrated and/or pumped concrete) with a minimum requirement of chemical admixtures.

#### 4.3.2. Concrete Mixtures Containing FRCA

The DRM-, EV- and PPM-designed recycled mixtures were proportioned with 100% FRCA replacement from the four FRCA sources previously mentioned. The target 28-day compressive strength was selected as 35 MPa as per CSA [60] for the C1 exposure class (i.e., structural concrete exposed to environmental conditions displaying freeze–thaw cycles and de-icing salts). Therefore, an air-entraining agent (AEA) was used in all mixtures to incorporate an air content of 7–8% required for the C1 exposure class as per CSA [55]. A water-to-cement ratio (*w*/*c*) of 0.35 was adopted for all FRCA mixes since preliminary trial batches indicated the need to reduce the *w*/*c* of the FRCA mixtures to yield close compressive strength results (i.e., 35 MPa) to companion conventional concrete mixes with *w*/*c* of 0.47. Moreover, the consistency of the mixture (i.e., slump values) was aimed to lie within 100 ± 20 mm. A summary of all mixtures is presented in Table 4. It worth noting that the FRCA material was added to the mixer in oven-dry conditions since previous studies [61,62] showed that pre-saturating FRCA would negatively affect the fresh- and hardened-state properties of the material.

The mixes proportioned using the DRM method (i.e., ACI—absolute volume method) treated the FRCA material as the same as natural fine aggregates, resulting in mixtures with high amounts of PC (e.g., nearly 500 kg/m^3^). Conversely, since the EV and PPM methods account for the RCP attached to the FRCA particles, the mixtures proportioned by these procedures presented a much lower PC content. In the case of the EV method, it was possible to obtain mixtures with Portland cement contents (≈373 kg/m^3^) very similar to those of conventional concrete designed with the ACI method (i.e., 370 kg/m^3^). Likewise, mixtures designed through the continuous PPM technique (distribution factor of 0.28) and with the use of inert fillers yielded final Portland cement contents ranging from 333 to 299 kg/m^3^ (using a minimum dosage of chemical admixtures).

### 4.4. Specimens’ Preparation and Testing Procedures

Forty litres of concrete were manufactured for each of the mixtures tested in this work to appraise their fresh-state behaviour. Furthermore, 100 mm × 200 mm concrete cylinders from all the fourteen mixes were fabricated according to CSA A23.2-3C [55], demoulded after 24 h and moist-cured over 28 days before testing.

### 4.5. Assessing Performance

#### 4.5.1. Fresh-State Behaviour

The fresh state behaviour of the fourteen mixtures was investigated by slump measurements and rheological characterization using a planetary rheometer (IBB). The IBB rheometer (Figure 4a) consists of an H-shape impeller (100 mm height and 130 mm length) and a bowl with a diameter of 360 mm and 250 mm height and is considered reliable when assessing concrete with consistencies ranging from 40 mm to 300 mm. The dataset (i.e., torque and rotation values) recorded by the IBB over the experimental process was used to plot the rheological profiles of all mixtures. It is worth noting that the IBB results are not displayed in fundamental units; thus, the yield stress and plastic viscosity are not given in Pa and Pa.s, respectively, but rather in N.m and rpm.

The rheological profiles of the mixtures evaluated were determined with a two-step process: an increase in shear rate up to approximately 43 RPM, followed by a decrease period at the same stepwise, keeping a constant rotation for roughly 10 s at each step (Figure 4b). The test was performed three consecutive times for each of the mixtures evaluated; the first test was verified to be largely different from the two consecutive ones and thus has been considered as pre-shearing. However, the second and third cycles were mainly identical; hence, to avoid “fluid memory” and mitigate potential segregation, the second cycle was selected for analysis and used in this research.

#### 4.5.2. Hardened-State Behaviour

The evaluation of hardened-state properties was performed through compressive strength and non-destructive tests. For compressive strength, three samples of each mixture were tested over time (i.e., 7, 14 and 28 days) as per CSA A23.1-14 [55]. Likewise, non-destructive techniques (i.e., surface electrical resistivity—ER and dynamic modulus of elasticity through ultrasonic pulse velocity test—UPV) were selected for this study to assess the microstructure quality of the distinct recycled concrete mixtures, as per ASTM C1876-19 [63] and ASTM C597-16 [64], respectively. It is worth mentioning that a viscoelastic gel was applied in between the transducers and concrete specimen to enable a suitable connection for the UPV test, and a frequency of 54 kHz was selected. Moreover, all specimens were tested in saturated surface-dry conditions at 23 ± 3 °C.

### 4.6. Microstructure Analysis

The microstructure quality of the concrete specimens was evaluated using scanning electron microscopy (SEM). The selected samples were first cut using a diamond-bladed masonry saw followed by subsequent polishing using a mechanical polishing table with grits of 30, 60, 140, 280 (80–100 µm), 600 (20–40 µm), 1200 (10–20 µm), 3000 (4–8 µm), 10,000 (2–4 µm) and 20,000 (1–2 µm). After that, the polished sections were dried at 60 °C for 24 h. Once the polishing process was completed, the samples were examined in a Jeol JSM- 6610LV SEM operated under a low vacuum, with an electron acceleration voltage of 20 kV and a spot size of 57. Backscattered electron (BSE) was conducted at different magnifications (i.e., ×300, ×500, ×750, ×1000 and ×1500) to study the mixture’s quality, morphology and distribution. The SEM images and the description of the microscopic features observed in the different recycled concrete mixtures are presented in the discussion section of the manuscript.

## 5. Experimental Results

### 5.1. Fresh-State Behaviour

The consistency results (i.e., slump test) of ACI, DRM, EV and PPM mixtures are presented in Figure 5. As previously mentioned, all mixtures assessed in the study were designed with a targeted slump range of 100 ± 20 mm. Overall, regardless of the mix design technique, incorporating the natural sand (NS) resulted in a slightly higher slump than MS mixtures (either for control or RCAs mixtures). The control ACI mixtures were manufactured with the same water content (174 kg/m^3^); yet, the ACI-NS resulted in a higher slump (110 mm) than ACI-MS (95 mm). Moreover, although the DRM mixtures were produced with the same water content as ACI mixtures, admixtures were required (i.e., 0.25% of mid-range/HRWR) to achieve the target slump (100 ± 20 mm). Otherwise, a lower amount of water (i.e., 131 kg/m^3^) was used for all EV mixtures along with a 0.32% of mid-range/HRWR. The PPM mixtures achieved the required slump with the lowest amount of water (i.e., ranging from 105 to 116 kg/m^3^) and a somewhat higher mid-range/HRWR (i.e., 0.32 to 0.40%) when compared to the EV mixtures. Finally, it is worth mentioning that mixtures containing FRCA-FG yielded overall higher slump values when compared to FRCA-CF.

The rheological profiles for the ACI, EV and PPM mixtures are presented in Figure 6. It is important to notice that due to the torque limitations of the IBB rheometer used for the analysis, the DRM mixtures could not be investigated in this section since they presented high cohesiveness and low flowability.

Analyzing the plots, one first verifies that all mixtures yielded a shear-thinning behaviour, i.e., decrease in viscosity (slope of the torque vs. rotation curve) as a function of the torque applied regardless of the mix design procedure adopted. Second, the PPM mixtures displayed the lowest yield stress, i.e., requiring a lower amount of torque to enable flow (i.e., 4.74 N.m to 8.17 N.m), followed by EV (7.32 N.m to 11.33 N.m) and ACI (10.70 N.m to 14.81 N.m)-mix-designed mixtures. Moreover, besides the rheological profiles and the minimum torque values mentioned above, two “extra” key rheological parameters can also be extracted (i.e., secant viscosity, the slope of the straight line between the first and the last deceleration points, and hysteresis area). A summary of all these values is displayed in Table 5. One may notice that ACI mixtures reached the lowest secant viscosity (7 N.m/rad/s, on average), followed by the PPM (23.45 N.m/rad/s) and EV (28.88 N.m/rad/s) mixtures, on average. Furthermore, all concrete mixtures fabricated incorporating NS aggregate exhibited greater secant viscosity (74% higher, on average). Likewise, comparing the crushing process adopted, the recycled mixtures made of FG aggregates yielded about 22% lower secant viscosity than CF mixtures; the only exception is the PPM-NS FG mixture, which yielded a secant viscosity around 37% higher than PPM-NS-CF.

Finally, the time dependency (thixotropic vs. rheopexy behaviour) of all mixtures was also evaluated by calculating the hysteresis area under the torque vs. rotation curve. Overall, all mixtures presented positive HA, corresponding to a thixotropic behaviour; in other words, the viscosity of those mixtures decreases over time under constant shear stress or shear rate. ACI mixes presented the lowest HA values, followed by PPM and EV. Furthermore, mixtures made of MS presented lower HA than those with NS, whereas mixtures made of FG aggregates also yielded lower HA values when compared to CF mixes.

### 5.2. Hardened-State Behaviour

In this section, compressive strength (CS) and non-destructive tests results are presented for all mixtures manufactured in the laboratory. Figure 7 displays the CS data obtained for the control and recycled concrete mixtures. Globally, regardless of the mix design method, concrete mixtures incorporating MS aggregate displayed slightly higher compressive strength (i.e., 5.1%) than NS-made mixtures. Moreover, the RCA production process significantly impacted the CS values once FG mixtures reached 23.5% higher than CF-made mixtures; this behaviour was even more evident for PPM-proportioned mixtures (i.e., PPM-NS-FG and PPM-MS-FG achieved 53% higher values than their respective CF mixtures).

Comparing the results, one verifies that the highest compressive strength values were obtained by PPM-proportioned mixtures (i.e., ranging from 41 to 65 MPa at 28 days), whereas the mixtures proportioned via DRM yielded the lowest results (i.e., between 19 and 21 MPa at 28 days). EV-designed mixes’ results lay between PPM and DRM mixtures (i.e., 26 to 31 MPa). The control mixes designed with natural (AC-NS) and manufactured (ACI-MS) sands presented 28-day compressive strength results within the range of 35 to 40 MPa.

The compressive strength results mentioned above correlate quite well with surface electrical resistivity (Figure 8a) and the dynamic modulus of elasticity (Figure 8b) values obtained. Analyzing the plots, one notices that the mixtures containing MS presented higher electrical resistivity (ER) values (5.9%, on average) than NS-made mixtures; likewise, the FG mixtures yielded 17.2% higher ER values, on average than those made of CF aggregates. Regarding the mix design techniques, the highest values were obtained for PPM mixtures (ER values between 10.8 and 14.3 KΩ∙cm at 28 days), followed by EV-proportioned mixes (ranging from 6.5 to 7.3 KΩ∙cm) and then DRM mixes (3.9–4.4 KΩ∙cm), respectively. A somewhat similar trend to the ER values was verified when evaluating the dynamic modulus of elasticity results of the RCA mixtures. In other words, incorporating MS and NS did not significantly impact the test results. Yet, the crushing process of the RCA significantly influenced the dynamic modulus of elasticity values since concrete specimens made of FG aggregates achieved a 16.3% higher dynamic modulus than CF-made mixtures. Moreover, PPM-proportioned mixtures presented the overall highest results (i.e., an average of 51 GPa for PPM-FG and 42 GPa for PPM-CF at 28 days) followed by EV mixes (i.e., 35 GPa for EV-FG and 27 GPa for EV-CF at 28 days) and finally DRM mixtures (i.e., an average of 16 GPa for both DRM-FG and DRM-CF, at 28 days). On the other hand, conventional concrete mixtures (ACI either NS or MS) obtained dynamic modulus of elasticity values of 49 GPa, just below the PPM-FG mixtures.

## 6. Analysis and Discussion

As previously mentioned, the present work is divided into two main parts: (1) production and characterization of FRCA materials and (2) appraisal of the fresh- and hardened-state properties of FRCA concrete mixtures proportioned using distinct mix design techniques. Although the data from the characterization of the FRCA, such as RCP content, specific gravity and water absorption and particle size distribution, were displayed in the experimental program section (since they were needed to design and manufacture the various recycled concrete mixtures), a global discussion on the obtained data seems appropriate herein. Therefore, the current *analysis and discussion section* will be divided into six sub-sections. First, a discussion of the influence of the raw materials and manufacturing process on the FRCA features is conducted. Second, a volumetric fraction evaluation of the various recycled mixtures components obtained through distinct mix design techniques is performed, followed by their impact on those mixtures’ fresh and hardened states. Finally, microstructure analyses of the recycled mixtures are conducted, along with a discussion on the binder efficiency of FRCA mixtures proportioned using distinct mix design techniques.

### 6.1. Influence of the Raw Materials and Manufacturing Process on the FRCA Features

The analysis of the characterization results demonstrates an important role of the raw materials used (i.e., NS or MS) or the RCA’s manufacturing processes (i.e., CF or FG) in the fresh and hardened states of FRCA concrete. Amongst the distinct properties appraised (i.e., specific gravity and water absorption, particle size distribution and RCP content), the RCP content showed an important influence of the raw material since FRCA made of MS presented a higher RCP than FRCA made of NS (i.e., 9% higher). This result is likely due to the rougher texture and more angular surface of the MS material, which may enhance the bond between the cement paste and aggregates. Likewise, comparing the FRCA’s manufacturing processes, CF-made materials yielded 23% higher fineness modulus and 30% more RCP attached (on average), which is linked to the number of crushing processes, i.e., the higher the number of crushing cycles, the lower the amount of RCP attached to the fine aggregate particles. Consequently, FRCA-CF presented 4% lower specific gravity and 20% higher water absorption than FRCA manufactured through the FG crushing process.

From the particle size distribution of the FRCAs, it was verified that a higher number of crushing cycles (i.e., FG process) produces a greater amount of fine material (i.e., particles smaller than 150 µm that are often discarded due to lack of quality). In other words, an appropriate and optimized process is required to produce FRCA to minimize waste generation (i.e., particles smaller than 150 µm) along with energy consumption during the manufacturing process. Otherwise, FG manufacturing processes generate FRCAs with 29.1% less RCP content (on average) than CF. Moreover, the greater the amount of RCP adhered to the particles, the lower the quality of the materials, and the more challenging it is for FRCA concrete mixture proportioning to achieve comparable fresh- and hardened-state properties to conventional companion mixtures.

The above discussion indicates that high amounts of RCP adhered to the FRCA particles may lead to a lower-quality material since more inner deterioration such as microcracks and flaws is expected. Conversely, the presence of RCP, depending on the condition of the cement grains (i.e., fully hydrated or not), may stimulate further hydration, which could help seal pre-existing inner cracks and thus enhance the overall quality of the FRCA material. In this regard, isothermal calorimetry was performed to quantify the flow and the amount of the heat released from the FRCA materials as mentioned in Section 4.2.1. Figure 9a,b present the heat flow curves and the cumulative heat curves (respectively) of distinct original FRCA materials (FRCA NS and FRCA MS).

The calorimetric measurements indicate that the overall acceleration of exothermic reactions from FRCAs was strongly reduced compared to general Portland GU-type cement samples. The FRCAs’ heat flow curves are not very well defined. Yet, two phases were identified (i.e., strong initial period and very slow induction period), which is a very different behaviour compared to the traditional GU cement behaviour with clear initial induction, accelerating and decelerating periods. Moreover, the results demonstrate that the difference in raw materials (i.e., MS versus NS) does not significantly influence the amount of heat released over the first 92 h of the test. Yet, both FRCAs demonstrated some hydraulic activity. The cumulative heat achieved by the FRCA materials after 92 h corresponded to about 15% of the heat released by the GU-type cement. This behaviour dos not significantly change the mechanical properties of the FRCA concrete over time, but this demonstrates that further hydrates might be theoretically formed in the system, which could contribute to an increase in the overall quality of the FRCA concrete. Further analysis is still required in future projects to quantify this potential aspect more precisely.

### 6.2. Volumetric Evaluation of Distinct Mix Design Methods to Proportion FRCA Mixtures

The distinct mix design methods used in this research played a very important role in the volumetric fraction components of both FRCA and CC mixes, as expected; yet, to better visualize this influence, Figure 10a,b are presented. The air percentage in the plots represents the fresh-state air entrainment/entrapment measured in the laboratory. Analyzing Figure 10, one notices that the overall amount of Portland cement in both conventional concrete mixtures made of NS and MS represents 11.8% of their total volume, whereas for DRM, EV and PPM-proportioned mixtures, the values obtained were 15.9%, 11.8% and 10.5% (on average), respectively. These values demonstrate the inefficiency of DRM techniques to proportion eco-efficient FRCA concrete to achieve comparable mechanical properties to CC. Conversely, the efficiency of PPM methods is conducive to the design of eco-efficient recycled mixtures, especially with the use of inert fillers. Furthermore, the type of FRCA manufacturing process and material (i.e., CF or FG, made of NS or MS) seems to play a role in the binder demand of the FRCA mixtures, since the higher the RCP content, the lower the binder demand, except for DRM mixtures, where no influence was noticed. Finally, it is worth noting that the volume of RCP is lower in mixtures containing FG aggregates when compared to CF. In contrast, the volume of residual (i.e., old) fine aggregate (presented as FRCA in Figure 10) is constant in mixtures proportioned with the same mix design technique (e.g., DRM-NS-CF has equal FRCA and higher RCP than DRM-NS-FG).

### 6.3. Fresh-State Behaviour

All mixtures assessed in the study were designed with a targeted slump range of 100 ± 20 mm, being thus considered (in practice) as “workable” mixtures to be used as cast-in-place/vibrated concrete. Yet, although similar consistencies (i.e., slump value) were selected for all mixtures, quite variable minimum torque results, ranging from 4.74 N.m (PPM-NS-FG) to 14.81 N.m (ACI-MS), were obtained. In general, the higher the slump value, the lower the minimum torque required (Figure 11). To attest to the validity of the minimum torque results gathered using the IBB rheometer, an analytical evaluation was conducted as proposed by [65] (Figure 11—τ_c_). These authors developed an analytical model to determine concrete mixtures’ yield stress (τ_c_) based on the slump test results. Analyzing the data below and the experimental comparison between slump vs. minimum torque, the calculated yield stress (τ_c)_ also decreases with the increase in a slump, which attests to the validity of the minimum torque results gathered.

According to Bingham’s model, the rheological behaviour of conventional concrete mixtures is described very often [66,67] and is represented by a linear relationship between shear stress and shear rate. However, granular suspensions such as concrete can also display different rheological behaviours, as the viscosity changes according to the torque applied. Moreover, the FRCA concrete mixes evaluated in this research were found to follow a distinct nonlinear trend (i.e., shear-thinning behaviour) where the viscosity lessens with the increase in shear rate. Recent research evaluating the rheological profile of concrete with reduced PC content [52] and RCA mixtures [68] verified that such eco-efficient mixtures might be precisely described in the fresh state through the Herschel–Bulkley model (Equation (2)). Hence, Figure 12 illustrates the rheological profiles gathered from all a) EV and b) PPM mixtures in comparison to CC mixtures and modelled with Herschel-Bukley’s approach.
(2)$$\tau =\tau_{0}+k_{HB}{\dot{\gamma }}^{n}$$
where *τ* is the shear stress, *τ*_0_ is the yield stress, $\dot{\gamma }$ is the shear rate, *k_HB_* is the viscosity constant of Herschel–Bulkley, and n is the flow behaviour factor, which is *n* < 1 for suspensions presenting shear-thinning behaviour and *n* > 1 for shear-thickening ones [66]. It is worth noting that for the sake of the following analyses, the concrete mixtures appraised in this work were assumed to behave as incompressible and homogenous fluids. Moreover, segregation has not been observed during the rheological tests, thus disregarded herein.

Analyzing the data gathered through Herschel–Bukley’s model (Table 6) and plotting them over the results acquired in the laboratory for each mixture (Figure 12) enhanced the rheological outcomes obtained in this research once the coefficient of determination obtained were averagely equal to 0.99. Moreover, one can verify that all mixtures displayed low-flow behaviour factors (i.e., n < 1), except for the ACI-NS mixture, where the n factor close to 1 indicates an almost linear behaviour (i.e., no change in viscosity as a function of torque). This correlation with the flow behaviour factor validates the shear-thinning aspect of the FRCA mixes since they require less torque applied in the system at higher shear stress regimes. The overall experimental initial torque required by each mix design represents 12.8 N∙m, 9.1 N∙m and 6.3 N∙m on average for CC, EV and PPM mixes, respectively. Furthermore, the k_HB_ constant was also observed to yield the lowest values for PPM mixes compared to EV and CC mixtures. It is important to notice that the higher k_HB_ values, the higher the slope of the rheological profile curve (i.e., higher viscosity).

The above results suggest that the PPM-proportioned mixtures have the most suitable rheological behaviour for vibrated and or pumped applications. Moreover, following the study on the minimum torque, the derivative of the Herschel–Bulkley model of each concrete mixture (Table 6) was performed to calculate the real viscosity of each mixture at different shear rates (Figure 13). The results show that, as previously discussed, ACI-NS is the only mixture where the viscosity did not change as a function of the torque applied (i.e., viscosity approximately constant and equal to 8 N∙m/rad/s). The higher the torque, the lower the viscosity for all the other mixtures.

### 6.4. Hardened-State Behaviour

The type of FRCA material (i.e., CF vs. FG) was found to have a low impact on the surface electrical resistivity (ER) and compressive strength (CS) of DRM and EV mixes and a more significant influence on the PPM-mix-proportioned mixtures. It is well known that the mechanical properties of concrete are directly related to the quality of its microstructure. Moreover, the results of CS and ER are quite well correlated; overall, the higher the compressive strength of the concrete, the higher the electrical resistivity. In general, the mixtures proportioned with FRCA-FG aggregates showed higher strength and ER results for the same *w*/*c* ratio; these results are due to the lower amount of RCP adhered to the FRCA-FG particles along with their more angular in shape and rougher in texture characters. Moreover, an increase of nearly 53% in CS and 31% in ER was obtained for PPM-designed mixes, whereas a slight average increase of about 16% and 6% was observed for EV and DRM mixes, respectively. The above results suggest that the FRCA particles’ features (inner quality, shape and texture) are more important in mixtures with a more optimized particle size distribution (improved granular skeleton) and thus have lower porosity. Conversely, the raw-material type (i.e., NS or MS) does not seem to influence the mechanical properties of FRCA concrete significantly.

Another important discussion to be addressed is on the dynamic elastic modulus (DEM) obtained in this research. The DEM results for the FRCA concrete mixtures follow a close correlation with the compressive strength, i.e., the higher the CS, the higher the DEM values. However, only the PPM-NS-FG and PPM-MS-FG mixtures could reach similar values to the companion conventional concrete mixtures (i.e., ACI-NS and ACI-MS). These findings may be justified by a thorough evaluation of the FRCA mixtures compositions. For instance, the replacement of natural fine aggregate by FRCA introduced theoretically more interfacial transition zones (ITZ) to the system, which may have governed (and lessened) the overall DEM of the recycled mixture. On the other hand, PPM-mix-designed recycled mixtures presented a higher DEM; this is associated with a better microstructure quality (i.e., optimization of the granular skeleton, lower porosity, etc.) of PPMs systems, enhancing the stiffness response of concrete mixtures as previously reported by [52,69]. Moreover, the FRCA manufacturing processes also impacted on the DEM results (i.e., on average FRCA-FG displayed a 16% greater DEM than FRCA-CF). The latter is related to the amount of RCP in the FRCA, which introduces more porosity and defects/flaws to the material’s microstructure and thus slows the ultrasonic propagation through the material.

The above discussion confirms both the influence of the manufacturing process (i.e., FG displays less RCP when compared to CF) and the mix-proportioning technique (i.e., PPM mixes showed improved behaviour compared to DRM and EV mixtures) on the mechanical properties of FRCA concrete.

### 6.5. Microstructure Analysis of FRCA Mixes

It is well established that the mechanical behaviour of CC depends on the quality of its compounds, such as hydrated products and the ITZ, which in turn is directly related to the porosity (or *w*/*c* ratio) of the mixture [70]. As previously discussed, FRCA is a multi-phase material comprised of OVA and RCP; hence FRCA concrete is expected to display a higher amount (and distinct types) of ITZs than conventional concrete (i.e., between OVA and RCP—old ITZ, and between FRCA and new cement paste—new ITZ) [71,72]. To further understand the hardened-state responses gathered in this research, the microstructure of the various recycled mixes proportioned using distinct mix design methods (i.e., DRM, EV and PPMs) were qualitatively analyzed through the scanning electron microscope (SEM). Thus, three SEM samples per mix were saw cut, polished and prepared as per 4.6. Figure 14 illustrates the SEM observations made at 750× magnification for FRCA mixtures incorporating NS.

The micrographs revealed that PPM-designed mixtures (i.e., both PPM-FG and PPM-CF) presented a much-enhanced microstructure when compared to EV or DRM proportioned mixes since suitable (and better) adherence between the residual cement paste (RCP) and the new cement paste (CP) was observed in all samples. Moreover, minor microcracks were identified within the RCP in both PPM-proportioned mixtures (highlighted areas in Figure 14a,b), being slightly more pronounced for the CF mix. The latter agrees with the mechanical results obtained in this work (i.e., PPM made of FG aggregates yielded better mechanical responses). These observations emphasize the suitability of PPMs to proportion FRCA concrete with an enhanced microstructure and mechanical response. Finally, the inclusion of inert fillers in PPM systems (i.e., limestone fillers with smaller particle size distribution than Portland cement) might also have contributed to the improvement of the new ITZ (Figure 14a,b) as previously reported [73,74,75], hence, enhancing the mechanical properties of FRCA concrete as discussed in Section 5.2.

Figure 14c,d indicate that EV and DRM-mix-designed mixtures display lower quality in ITZs (i.e., old and new) than PPM mixes. Yet, EV samples showed an enhanced quality of their new cement paste compared to DRM mixes (Figure 14c,d), which agrees with their better mechanical responses, especially compressive strength. Moreover, it has been found that the quality of the new ITZ and cement paste of FRCA concrete is influenced by the material’s pre-saturation, which is released over time to the new cement paste and generates further porosity in the mix [54]. The latter occurs especially in recycled mixtures that disregard the RCP in the mix design, such as DRM, and are indeed noticed in Figure 14d. However, it is worth mentioning that recent research [76] performed on cementitious mortars made of FRCA shows that the water may be transferred within the first 6 min and remains constant for up to 90 min. Therefore, further investigation is still required to better understand water transfer in cementitious materials made of FRCA since this behaviour may depend on the recycled material’s features such as type (i.e., mineralogy), shape and texture and porosity and absorption of the aggregate, along with the amount and quality of the residual cement paste.

### 6.6. Binder Efficiency

In order to evaluate the eco-efficiency of conventional concrete mixtures, Damineli et al. [77] developed an index, the so-called binder intensity (bi), which accounts for the amount of binder (kg/m^3^) required to obtain one unit of a given mechanical property; for example, 1 MPa of compressive strength at 28 days. Figure 15a displays the amount of new Portland cement (presented in Table 4) used to reach the 28-day compressive strength results gathered for each of the three mix design methods utilized in this work. Analyzing the plot, it is clear that the overall performance of recycled mixtures is not directly linked to the amount of Portland cement used but rather to the microstructure quality of the material (related to the efficiency of the mix-proportioning technique). For instance, among all FRCA mixes, the PPM-MS-FG yielded the highest 28-day compressive strength (i.e., nearly 65 MPa) while presenting a moderate binder content (332 kg/m^3^).

To evaluate the binder efficiency of all conventional concrete and FRCA mixtures studied in this experimental program, the *bi* factor was calculated for all of them, and the results are illustrated in Figure 15b. It can be observed from the plot that PPM-mix-designed mixtures (green triangles) were the most eco-efficient FRCA mixes designed in this research since they yielded the lowest *bi* values (8.3 kg∙m^3^∙MPa^−1^ with an approximate average 28-day compressive strength of 53 MPa). Furthermore, one may notice that some PPM mixtures are quite close to the bottom line of the plot (i.e., Portland cement content = 250 kg/m^3^), which represents the lowest amount of Portland cement conventionally used in the market for conventional concrete. Otherwise, conventional concrete mixes presented an overall *bi* factor of 10 kg∙m^3^∙MPa^−1^ for an average 28-day compressive strength of about 37 MPa, while DRM and EV mixes displayed *bi* factors of approximately 25 and 13.1 kg∙m^3^∙MPa^−1^ for average compressive strengths of 20 and 29 MPa, respectively. These results demonstrate the promising character of PPM techniques in creating mix proportions of eco-efficient FRCA concrete with suitable fresh- and hardened-state properties.

## 7. Conclusions

The current research aimed to understand the overall behaviour of FRCA concrete made of distinct raw materials (i.e., natural sand—NS or manufactured sand—MS), manufacturing processes (i.e., CF vs. FG) and mix design techniques (DRM, EV and PPMs). The main findings of this research are presented hereafter:It is clear that the manufacturing (i.e., crushing, sieving, etc.) process influences the overall quality of the FRCA material. With only two consecutive series of crushing, the so-called CF generates recycled particles with coarser PSD, and high amounts of adhered RCP. Yet, multiple crushing series, the so-called FG, were observed to produce FRCA particles with similar PSD to natural aggregates and displayed less RCP attached to the particles due to further separation of mortar during crushing. Moreover, CF particles seem to be more rounded in shape and smoother in texture when compared to FG particles.The fresh-state behaviour of FRCA mixtures proportioned through both EV and PPM methods demonstrated an interesting performance, showing a shear-thinning profile (i.e., a decrease in viscosity as a function of the torque). Yet, PPM-mix-designed FRCA concrete demonstrated the best rheological variables (i.e., minimum torque required and lowest secant viscosity) among all mixtures; the latter indicates its suitability for applications under high-torque regimes such as pumping and or vibrating processesInteresting hardened-state properties (i.e., compressive strength and dynamic elastic modulus) were achieved using both EV and PPM techniques. However, in general, the DRM method yielded the worst mechanical performance obtained in this work. Yet, PPM-mix-designed FRCA proved to be the most promising mix design technique to proportion FRCA concrete mixes with suitable mechanical properties, binder efficiency and an enhanced microstructure (i.e., SEM analysis).It seems that both the aggregate types (NS vs. MS) and crushing processes (CF vs. FG) play an important role in the overall FRCA quality (i.e., RCP content, particle shape, texture, etc.) along with fresh- and hardened-state performances. Yet, the results obtained in this experimental campaign demonstrated that the mix design procedure adopted to produce FRCA concrete is even more important than the FRCA features and inner quality.Finally, microscopic analysis needs to be assessed considering distinct FRCA concrete mixes through the use of both EV and PPM methods in order to investigate the different failure mechanisms of FRCA concrete, especially when subjected to harsh environmental conditions. This would propose valuable remarks on the mechanical and durability-related properties of FRCA mixes against CC mixes as well as potential improvements on the EV mixes.

## Figures and Tables

**Figure 1 materials-15-01355-f001:**
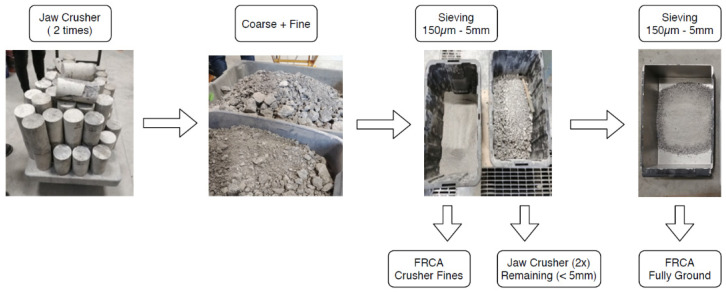
Summary of crusher fines (CF) and fully ground (FG) method of production.

**Figure 2 materials-15-01355-f002:**
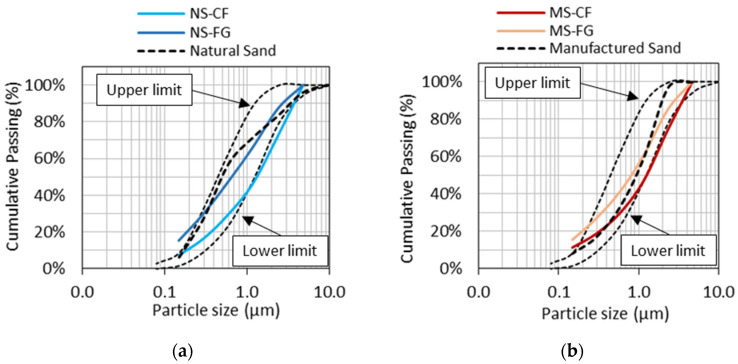
Particle size distribution curves for FRCA derived from natural (**a**) and manufactured sand (**b**) companion conventional concrete.

**Figure 3 materials-15-01355-f003:**
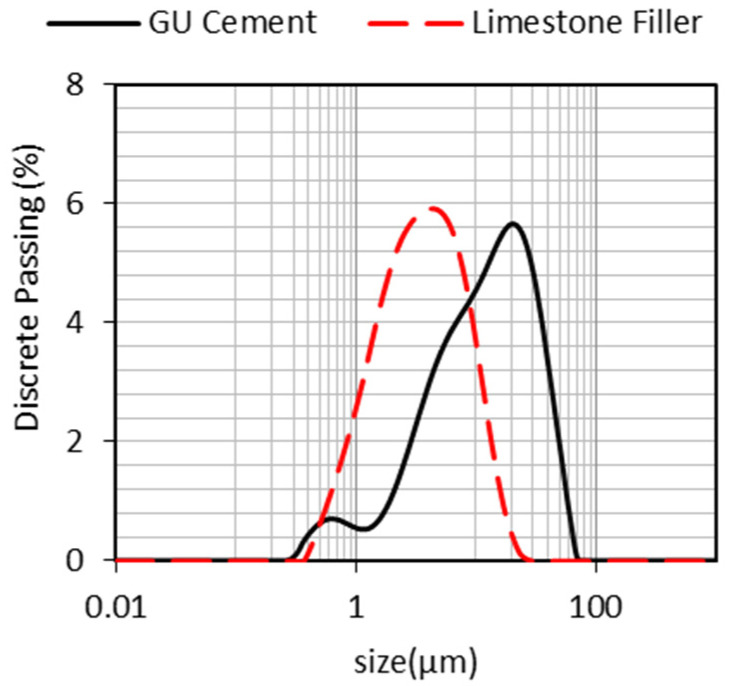
PSD of PC and limestone filler used in the PPM FRCA mixtures.

**Figure 4 materials-15-01355-f004:**
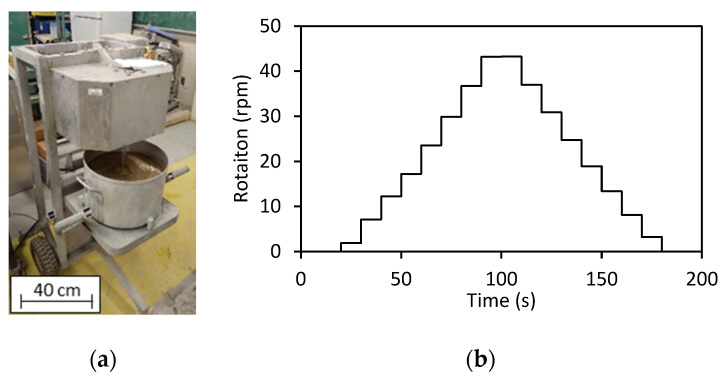
(**a**) IBB rheometer and (**b**) test programme used for analyzing the rheology cycle.

**Figure 5 materials-15-01355-f005:**
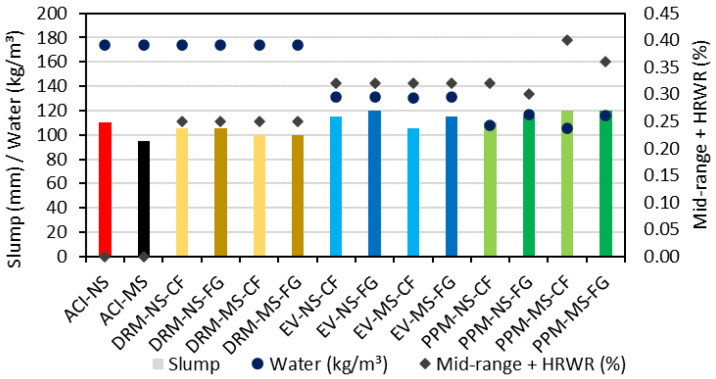
Comparison between slump, water content and admixtures of distinct mixtures.

**Figure 6 materials-15-01355-f006:**
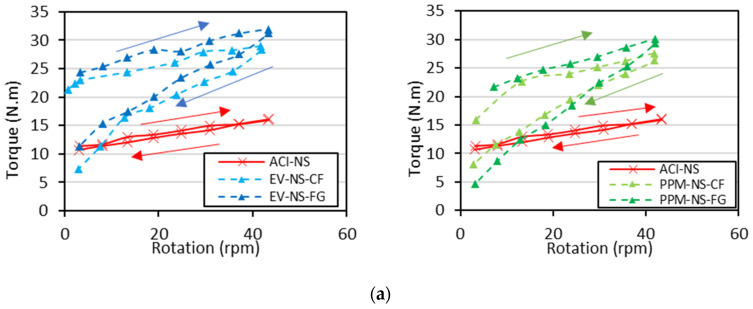
Rheological profiles of concrete mixtures made of NS (**a**) or MS (**b**).

**Figure 7 materials-15-01355-f007:**
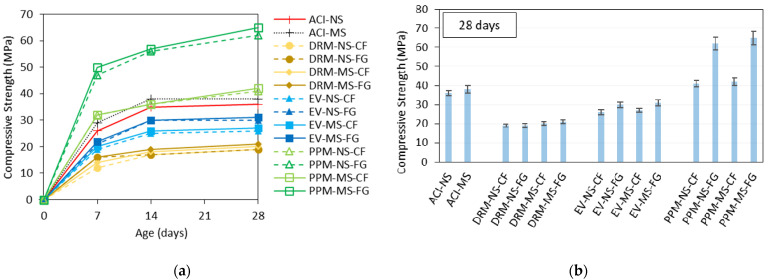
Compressive strength for all mixtures appraised: (**a**) over time and (**b**) at 28 days with the error bars.

**Figure 8 materials-15-01355-f008:**
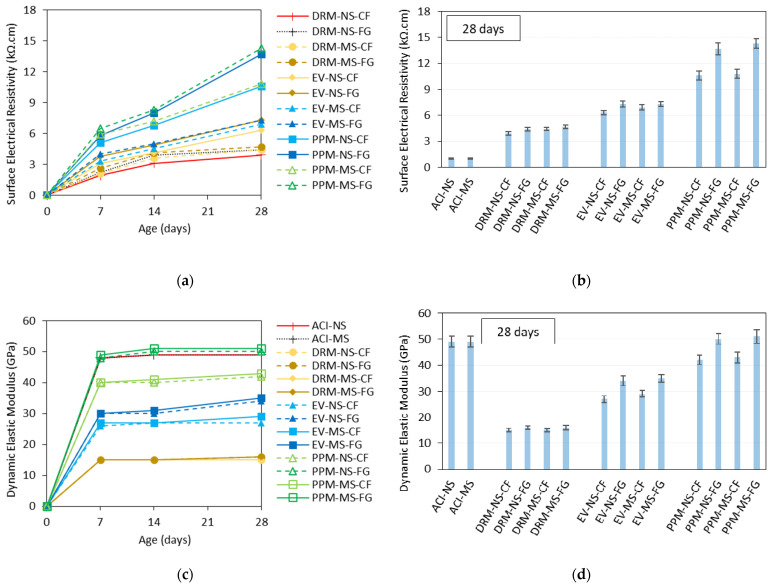
Non-destructive tests results: (**a**) surface electrical resistivity over time, (**b**) surface electrical resistivity at 28 with error bars, (**c**) dynamic elastic modulus over time, and (**d**) dynamic elastic modulus at 28 with error bars.

**Figure 9 materials-15-01355-f009:**
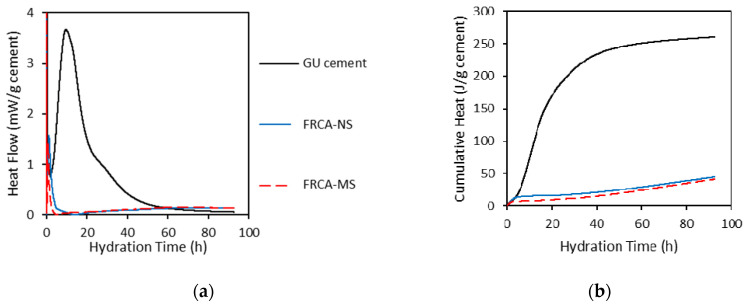
Isothermal calorimetry analysis: (**a**) heat flow curves and (**b**) cumulative calorimetric curves of original and powder FRCA samples.

**Figure 10 materials-15-01355-f010:**
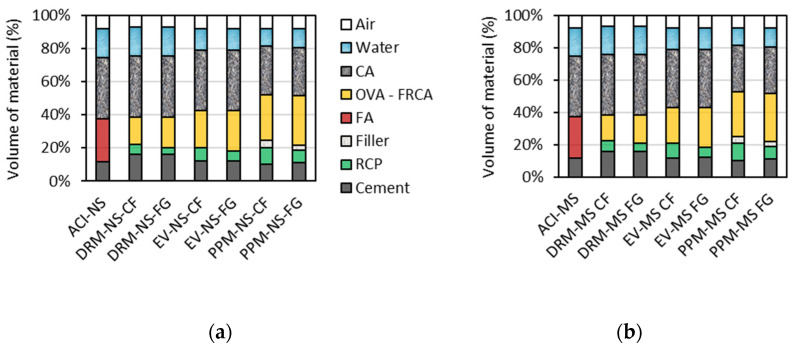
Volumetric fraction of each component for mixtures with FRCA made of NS (**a**) or MS (**b**).

**Figure 11 materials-15-01355-f011:**
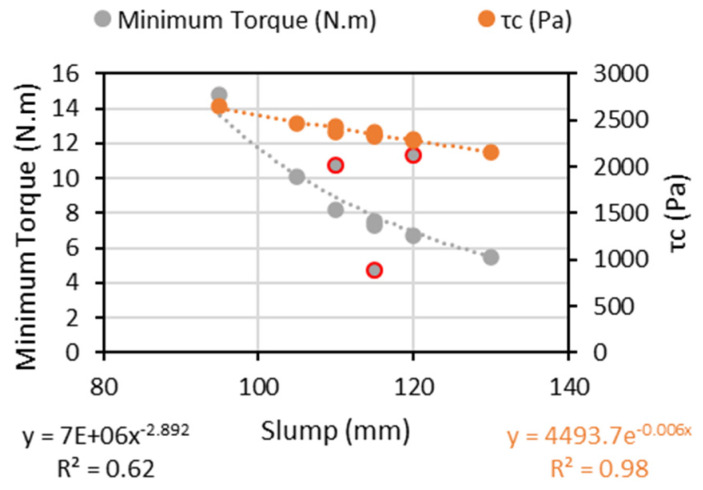
Yield stress experimental versus calculated through Roussel model.

**Figure 12 materials-15-01355-f012:**
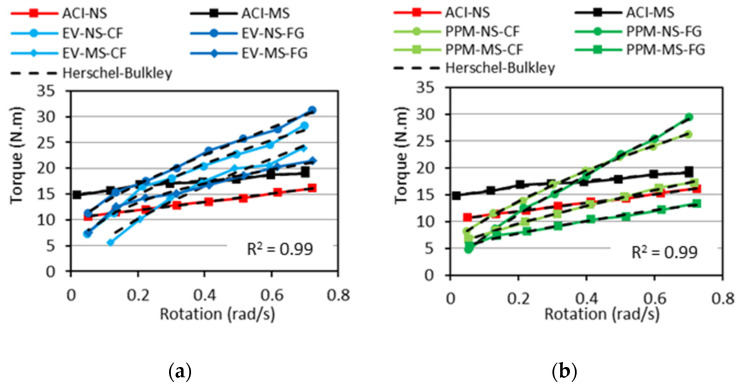
Rheological profiles of (**a**) EV and (**b**) PPM mixes compared to Herschel–Bukley’s model.

**Figure 13 materials-15-01355-f013:**
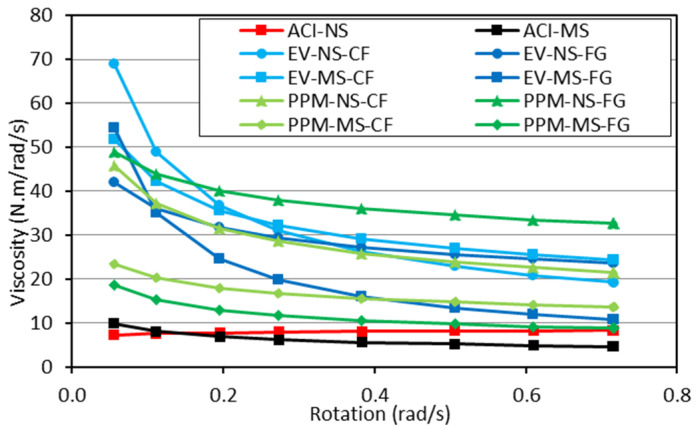
Real viscosity of non-Newtonian mixtures calculated through Herschel–Bulkley model.

**Figure 14 materials-15-01355-f014:**
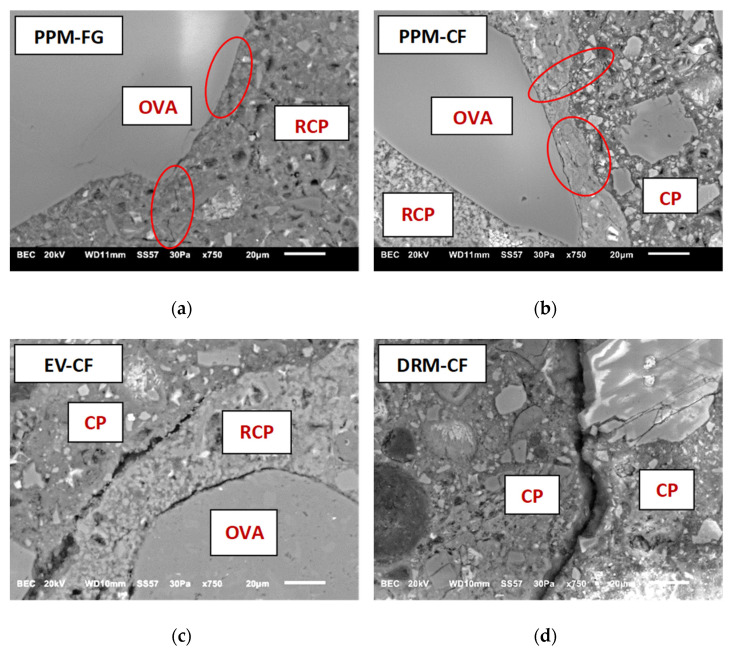
SEM micrographs of the old and new ITZs present in different RCA mixtures designed by (**a**) PPM-NS-CF, (**b**) PPM-NS-FG, (**c**) EV-NS-CF and (**d**) DRM-NS-CF—BSE images at a magnification of 750×.

**Figure 15 materials-15-01355-f015:**
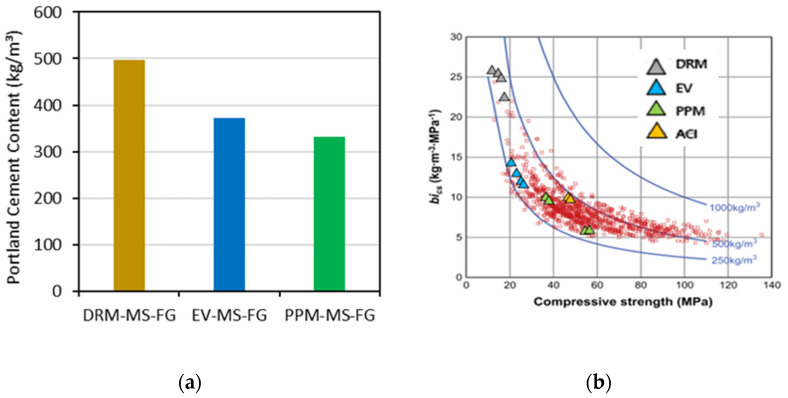
(**a**) Portland cement (PC) content levels for the mixtures in analysis; (**b**) relationship between bi and compressive strength at 28 days (adapted from [77]).

**Table 1 materials-15-01355-t001:** Companion conventional concrete mix design proportions.

ACI-NS	Type	Mass (kg/m^3^)
Cement	GU	370
Fine aggregate	Natural sand	898
Coarse aggregate	Limestone	1032
Water	-	174
ACI-MS		Mass (kg/m^3^)
Cement	GU	370
Fine aggregate (MS)	Manufactured sand	934
Coarse aggregate	Limestone	1032
Water	-	174

**Table 2 materials-15-01355-t002:** Properties of FRCA and natural aggregates.

Physical Property	FRCA-NS-CF	FRCA-NS-FG	NS	FRCA-MS-CF	FRCA-MS-FG	MS
RCP content (wt.%)	15.5	11.5	-	16.8	11.4	-
Saturated surface-dry specific gravity (kg/L)	2.47	2.56	2.70	2.51	2.58	2.76
Oven-dry specific gravity (kg/L)	2.29	2.41	2.67	2.33	2.44	2.74
Water absorption (%)	7.87	6.38	0.86	7.76	6.16	0.65

**Table 3 materials-15-01355-t003:** Physicochemical characterization of GU cement and limestone filler.

Chemical Composition	Filler	Cement	Mineralogical Phases		
CaO	52.28	61.4	C_3_S	52.9	
SiO_2_	4.09	18.9	C_2_S	14.3	
Al_2_O_3_	0.23	4.9	C_3_A	6.7	
Fe_2_O_3_	0.15	3.7	C_4_AF	11.3	
MgO	1.69	2.3	Physical Properties	Cement	Filler
K_2_O	0.07	0.9			
Na_2_O	0.04	0.3	Specific Gravity (g/cm^3^)	3.03	1.00
SO_3_	-	5.3			
LOI	41.3	2.1	SSA (m^2^/g)	2.60	3.70

**Table 4 materials-15-01355-t004:** FRCA and control concrete mix design proportions.

Mixture	Cement ^b^ (kg/m^3^)	FRCA (kg/m^3^)	Natural FA (kg/m^3^)	Natural CA (kg/m^3^)	Filler (kg/m^3^)	Water (kg/m^3^)	Mid-Range + HRWR ^a^ (%)
ACI-NS	370	-	685	1032	-	174	-
ACI-MS	370	-	704	1032	-	174	-
DRM-NS-CF	497	524		1032	-	174	0.25
DRM-NS-FG	497	546		1032	-	174	0.25
DRM-MS-CF	497	533		1032	-	174	0.25
DRM-MS-FG	497	551		1032	-	174	0.25
EV-NS-CF	373	714	-	1005	-	131	0.32
EV-NS-FG	373	740	-	1014	-	131	0.32
EV-MS-CF	373	732	-	1004	-	131	0.32
EV-MS-FG	373	752	-	1006	-	131	0.32
PPM-NS-CF	308	879	-	806	108	108	0.32
PPM-NS-FG	333	907	-	797	83	117	0.30
PPM-MS-CF	299	898	-	809	118	105	0.40
PPM-MS-FG	332	915	-	798	84	116	0.36

^a^ HRWR: high-range water-reducer admixture. ^b^ Cement content presented in Table 4 refers to the total amount of new cement. The old cement (i.e., RCP) is accounted for in the FRCA portion.

**Table 5 materials-15-01355-t005:** Fresh-state properties of concrete mixtures appraised.

Mixture	ACI-NS	EV-NS-CF	EV-NS-FG	PPM-NS-CF	PPM-NS-FG
Minimum torque (N.m)	10.70	7.33	11.33	8.17	4.74
Secant viscosity (N.m/rad/s)	8.05	32.28	29.73	27.90	38.10
Hysteresis area (N.m.rad/s)	0.59	5.39	4.98	3.63	5.00
**Mixture**	**ACI-MS**	**EV-MS-CF**	**EV-MS-FG**	**PPM-MS-CF**	**PPM-MS-FG**
Minimum torque (N.m)	14.81	10.12	7.48	6.75	5.45
Secant viscosity (N.m/rad/s)	6.15	31.67	20.83	15.88	11.91
Hysteresis area (N.m.rad/s)	0.36	4.66	4.09	3.09	3.19

**Table 6 materials-15-01355-t006:** Parameters of Herschel–Bukley’s equation and minimum torque required for all mixtures inves-tigated.

Mixture	Minimum Torque—Experimental (N∙m)	τ_0_ (N∙m)	K_HB_ (N∙m/rad/s)	*n*
ACI-NS	10.7	10.4	8.0	1.05
ACI-MS	14.8	14.5	6.0	0.71
EV-NS-CF	7.3	0.2	32.5	0.50
EV-NS-FG	11.3	8.8	28.3	0.78
EV-MS-CF	10.1	0.1	31.2	0.70
EV-MS-FG	7.5	0.1	23.8	0.37
PPM-NS-CF	8.2	4.9	27.7	0.71
PPM-NS-FG	4.7	1.8	36.8	0.84
PPM-MS-CF	6.7	5.1	16.2	0.79
PPM-MS-FG	5.5	4.2	11.3	0.70

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
