# Peer review of "Influence of the Mix Proportion and Aggregate Features on the Performance of Eco-Efficient Fine Recycled Concrete Aggregate Mixtures"

_materials, 2022, doi:10.3390/ma15041355_

Round 1

Reviewer 1 Report

Titles:

It is necessary that the title specify the origin of the recycled aggregate (concrete, ceramic, glass, etc.). It is recommended to improve it; in the same way, it is considered very long, to simplify.

Abstract:

It is necessary to specify improves the way in which this research tries to make behavioral deductions (variables, tests performed, etc.)

It is necessary to present the most significant results in quantitative terms.

Keywords:

It is necessary to specify better: mix-design technique of mortar, hardened property of mortar

Introduction:

It is considered that the state of knowledge of both recycled coarse aggregates from concrete and fine aggregates from concrete is not sufficient (it is poor or lacks all the information). It is necessary that the authors carry out a search in databases such as Scopus, Compendex or WoS of the last years. It is strange that the most recognized researchers on these topics do not appear, such as:

Corinaldesi, V.

Moriconi, G.

Cabrera-Covarrubias, F.

Etc.

Techniques for dosing:

It is considered that the information presented needs to be simplified. It is not necessary to repeat the information contained in other previous works (definition, hypothesis, etc.). Instead, a critical and comparative reflection of existing methods must be carried out.

Materials and Method:

Justify all the variables used (ideal); for example: jaw crusher (opening), number of times crushing, etc.

Characterization:

It is necessary to include the make and model of all equipment used in the tests.

Figure 2. Include the maximum and minimum limits of regulations; for example, ASTM

Lines 300 and 333, reference error.

Justify studying only 100% replacement (not usual in research or applications)

Figure 4a): Include graphic scale

It is necessary to indicate the standard that was used in each trial planned in this work.

Results:

Figures 7 and 8, include ranges of variation.

Why weren't more tests (mechanical and physical) done on the mortars? (traction, volumetric weight, porosity, etc. etc.) It is necessary to include a complete study of the different study results.

Why are microstructure results not presented in this section? (or all or none)

Discussion:

Unproven hypotheses are not accepted; for example: ... This result is likely to be ... (demonstrate, validate, or determine)

It is necessary that all the results be contrasted with previous studies by other researchers.

Conclusions:

Review after fixing previous comments.

Bibliography: See previous comments.

Reviewer 2 Report

The present paper deals with the use of fine recycled concrete aggregates to produce eco-efficient concrete material. The study is extensive in all its parts and the reported results are of interest for the Scientific community and fit with the aim of the Journal. Thus I suggest it for the publication after minor changes.

As already mentioned the paper is very long in all these parts that anyway sound coherent with the aim of the study. I suggest to arrange some information differently to help readers to follow the structure and the content of the paper. Especially in chapter 4 "Materials and methods", the Authors reported also the results on the characterization of the prepared FRCA. I suggest to move this information in the following chapter "Experimental results". In addition, all the plots reported to show the experimental results are small and this does not help the reader. Thus I suggest to enlarge the reported plots.

Reviewer 3 Report

This manuscript evaluates the overall performance of sustainable FRCA mixtures proportioned through distinct techniques (ie, direct replacement, EV and PPMs), and incorporating different types of aggregates (ie, natural and manufactured sand) and manufacturing processes (ie, crusher fines and fully ground ). In summary, the research is interesting and provides valuable results, but the current document has several weaknesses that must be strengthened in order to obtain a documentary result that is equal to the value of the publication.
General considerations:
(1) Concerning the presentation of the contents, the document is acceptable. Nonetheless, it is recommended that authors develop proofreading to avoid common mistakes such as word confusion (eg in table 3: “SiO2” instead of “SiO2”, “g/ cm3” instead of “g/cm3”, in table 4 “cementb” instead of “cement”), format error (lines 330 or 333), continuous repetition of the same words and expressions, font error (line 294), etc.
(2)The document contains a total of 66 employed references, of which 29 are publications produced in the last 5 years (44%), 20 in the last 5-10 years (30%), 17 than 10 years old (26% ), implying a total percentage of 74% recent references. In this way, the total number is sufficient, and their actuality is high. But the references should be checked carefully, for example [8] is missing the journal name.
(3) The authors cites a lot of papers from CBM but none from Materials.
Title, Abstract and Keywords:
(4) The abstract is complete and well-structured and explains the contents of the document very well. Nonetheless, the part relating to the results could provide numerical indicators obtained in the research.
(5)The research purpose is not refined enough, some explanatory text should not be reflected in the abstract, it is more appropriate to put it in the introduction.
Chapter 1: Introduction
(6)The introduction gives a too simple, and even incomplete, view of the problems related to your topic and should be revised and completed with citations to authority references (Combined effects of nano-silica and silica fume on the mechanical behavior of recycled aggregate concreteï¼›Fracture behavior of a sustainable material: Recycled concrete with waste crumb rubber subjected to elevated temperatures).
(7) The novelty of the study is not apparent enough. In the introduction section, please highlight the contribution of your work by placing it in context with the work that has done previously in the same domain.
(8)On a general level, the research on this content is reasonable, and the explanation of the objectives of the work may be valid. However, the limitations of your work are not rigorously assumed and justified.
Chapter 2: Background
(9) The background and the introduction are partly duplicated, and the logic of this part needs to be strengthened.
Chapter 3: Scope of work
(10) It is recommended that this part only requires the work of writing this article, and does not need to describe the work of the predecessors.
Chapter 4: Materials and Methods
(11) The materials characterization section should be classified as 4.2.
(12)The dispersion effect of the sample after the “sodium hexametaphosphate solution” treatment is not given in the article, so can the reliability of the CSA results be guaranteed?
(13)There are 5 curves in Figure 2, but only 3 are marked. It is recommended to mark the remaining two curves.
(14) The introduction of the previous EV method in 4.3.1 can be deleted, just write your own practice in this article.
Chapter 5: Experimental results
(15)The discussion of experimental results should be richer. There are more experimental groups in this article, so it is recommended to compare the data in multiple directions.
(16) Figure 7 can consider adding several compressive strength diagrams to avoid too many data curves and inconvenient comparison.
Chapter 6: Analysis and Discussion
(17) The analysis and discussion of the experimental results should be more complete, and the explanation of the mechanism should be more rigorous.
(18) The explanation of some reasons should be justified and evidenced, and supplementary experiments or references may be considered to prove.
Chapter 7: Conclusions
(19) After all that has been read, the content of this article has indeed contributed to the overall performance research of FRCA, but its limitations need to be verified before it can be effectively applied.
(20) It should mention the scope for further research as well as the implications/application of the study.

Round 2

Reviewer 1 Report

The authors have fixed the comments made. The work can be published.
Congratulations

Reviewer 3 Report

Congrats!

The authors have successfully addressed all my comments. Therefore, I recommend the publication of this manuscript.